# Bioenergetic Balance of Continuous Venovenous Hemofiltration, a Retrospective Analysis

**DOI:** 10.3390/nu14102112

**Published:** 2022-05-18

**Authors:** Joop Jonckheer, Alex Van Hoorn, Taku Oshima, Elisabeth De Waele

**Affiliations:** 1Department of Critical Care, Universitair Ziekenhuis Brussel, Laarbeeklaan 101, 1090 Jette, Belgium; alex.vanhoorn@uzbrussel.be; 2Emergency and Critical Care Medicine, Graduate School of Medicine, Chiba University, 1-8-1 Inohana Chuo-ku, Chiba City 260-8677, Japan; t_oshima@chiba-u.jp; 3Departement of Nutrition, Universitair Ziekenhuis Brussel, Laarbeeklaan 101, 1090 Jette, Belgium; elisabeth.dewaele@uzbrussel.be

**Keywords:** continuous renal replacement therapy, continuous venovenous hemofiltration, indirect calorimetry, resting energy expenditure, citrate, non-intentional calories

## Abstract

(1) Background: Nutrition therapy guided by indirect calorimetry (IC) is the gold standard and is associated with lower morbidity and mortality in critically ill patients. When performing IC during continuous venovenous hemofiltration (CVVH), the measured VCO_2_ should be corrected for the exchanged CO_2_ to calculate the ‘true’ Resting Energy Expenditure (REE). After the determination of the true REE, the caloric prescription should be adapted to the removal and addition of non-intentional calories due to citrate, glucose, and lactate in dialysis fluids to avoid over- and underfeeding. We aimed to evaluate this bioenergetic balance during CVVH and how nutrition therapy should be adapted. (2) Methods: This post hoc analysis evaluated citrate, glucose, and lactate exchange. Bioenergetic balances were calculated based on these values during three different CVVH settings: low dose with citrate, high dose with citrate, and low dose without citrate. The caloric load of these non-intentional calories during a CVVH-run was compared to the true REE. (3) Results: We included 19 CVVH-runs. The bioenergetic balance during the low dose with citrate was 498 ± 110 kcal/day (range 339 to 681 kcal/day) or 26 ± 9% (range 14 to 42%) of the true REE. During the high dose with citrate, it was 262 ± 222 kcal/day (range 56 to 262 kcal/day) or 17 ± 11% (range 7 to 32%) of the true REE. During the low dose without citrate, the bioenergetic balance was −189 ± 77 kcal/day (range −298 to −92 kcal/day) or −13 ± 8% (range −28 to −5%) of the true REE. (4) Conclusions: Different CVVH settings resulted in different bioenergetic balances ranging from −28% up to +42% of the true REE depending on the CVVH fluids chosen. When formulating a caloric prescription during CVVH, an individual approach considering the impact of these non-intentional calories is warranted.

## 1. Introduction

The individualization of medical nutrition by the assessment of resting energy expenditure (REE) with indirect calorimetry (IC) is associated with lower mortality in adult patients admitted to the intensive care unit [1]. Based on the measured REE, progressive nutrition to achieve normocaloric feeding between day 3 and 5 as compared to hypocaloric feeding, is associated with lower morbidity and mortality [2,3]. Overfeeding, especially in the first days of admission, is also deleterious and is associated with higher mortality [4,5]. The beneficial effects on disease evolution and outcome of IC-guided nutrition therapy can probably be extrapolated in other cohorts, including patients with acute kidney injury receiving renal replacement therapy [6,7]. Acute kidney injury, a frequent (prevalence of 13–78%) condition in the intensive care unit, leads to toxin accumulation and can be treated with renal replacement therapy [8]. Continuous renal replacement therapy (CRRT) is usually preferred over intermittent dialysis in critically ill patients as it is hemodynamically better tolerated, offers prompt correction of life-threatening metabolic imbalances, and allows for the adequate control of fluid balance [9]. Depending on the technique used for solute removal, dialysis is called continuous venovenous hemofiltration (CVVH) or continuous venovenous hemodiafiltration (CVVHDF). Citrate is frequently used as an anticoagulant for the extracorporeal circuit of CRRT [10]. Although citrate reaches high concentrations in the extracorporeal circuit, the concentration in the systemic circulation is much lower. Citrate is partially removed in the effluent during CRRT [10,11]. When the extracorporeal blood enters the systemic blood circulation, citrate dissolves in the total blood volume and is cleared by the liver resulting in very low plasma levels [12].

Due to the exchange of solutes during CRRT, nutrition therapy is more challenging. First, CRRT induces CO_2_ exchange outside the lungs, leading to inaccurate REE measurements with IC [13]. The earlier manuscript of this MECCIAS trial clarified how the CO_2_ is exchanged during CVVH and how a correction factor can be integrated into the REE measurements by IC to calculate the ‘true REE’ [14,15]. Second, non-intentional calories are exchanged during CRRT [7,16]. During CVVHDF, up to 1434 kcal/day were administered due to citrate, glucose, and lactate [11,12]. Not all dialysis fluids contain the same amount of glucose and lactate, which results in different bioenergetic balances (Table 1). Therefore, depending on the dialysis fluid used, different amounts of non-intentional calories are given to the patient [11,12]. Furthermore, due to the hepatic clearance, citrate use is contraindicated in liver function impairment. CRRT regimens have been developed with citrate-free fluids, which can affect the bioenergetic balance [10].

Finally, besides the different compositions of dialysis fluids, the flows used during CRRT can also impact the exchange of non-intentional calories. Dialysis fluid flows are tailored to the patient’s individual need, resulting in different exchange rates of non-intentional calories [10,11,12]. To our knowledge, no data exist on the relative proportion of non-intentional calories related to the true REE during CVVH.

In this study, we aimed to analyze the bioenergetic balance of different CVVH settings during the MECCIAS trial and compared it to the true REE to achieve general considerations and recommendations.

## 2. Materials and Methods

A retrospective analysis was performed on all patients included in the prospective MECCIAS trial (MEtabolic Consequences of Continuous venovenous hemofiltration on Indirect cAlorimetry) [15]. The prospective trial explored the influence of CO_2_ exchange on the REE of different settings of CVVH: during a standard of care low dose CVVH with citrate, during a high dose CVVH with citrate, and during a low dose CVVH without citrate. The low dose CVVH with citrate was performed according to a local low dose protocol with citrate predilution with a targeted effluent dose of 25 to 30 mL/kg/h. For the high dose CVVH setting, the postdilution fluid was intended to be increased so an effluent dose of 50 mL/kg/h was achieved. The CVVH was then adjusted again to the first low dose settings and the citrate predilution was replaced by normal saline (NaCl 0.9%). For this retrospective report, we included all of the CVVH runs with glucose and lactate measurements in effluent. Glucose and lactate were measured using a point of care blood gas analyzer (ABL 90 flex^®^, radiometer; Bronshoj, Denmark). The bioenergetic balance of the CVVH was defined as the net combined exchange of citrate, glucose and lactate converted into kcal per 24 h. The loss of glucose and lactate was calculated by multiplying the effluent dose with the measured glucose and lactate concentration of the fluid, respectively. Citrate in the extracorporeal circulation before the filter was calculated by multiplying the predilution flow with the citrate content of the fluid and dividing this by the flow of predilution plus the flow of blood [citrate]_ECC_ = [citrate]_pre_ × Q_pre_/(Q_pre_ + Q _blood_). The sieving coefficient of the citrate is approximatively 1 [17,18]. Therefore, the citrate concentration in the effluent is the same as in the blood after the predilution fluid is added. The removal was calculated by multiplying the theoretical concentration in effluent with the effluent flow [17]. The following calculations were performed, to convert the amount of the different carbohydrates that are lost in the effluent to energy loss: 1 g of citrate was considered to contain 2.5 kcal; 1 g of glucose was considered to contain 3.75 kcal; and 1 g of lactate was considered to contain 3.6 kcal [11,12]. The bioenergetic balance of the CVVH was compared with the true REE, as described in the MECCIAS trial. The true REE was measured and calculated during each different run, as different components of the CVVH will influence this measurement and calculation [15].

Statistical advice was sought at the statistical department of the Vrije Universiteit Brussel. Due to the low number of samples, normality was assumed to be not assessable. Due to missing data, an unpaired analysis was performed. A one-way ANOVA was performed to compare the multiple groups and a student *t*-test was performed to compare the two groups. *p* < 0.05 was considered a statistically significant difference. A Prism version 7.0c (GraphPad, San Diego, CA, USA) was used for the statistical analysis.

## 3. Results

Nineteen CVVH runs from nine different patients were included for the post hoc analysis as they contained a blood gas analysis of the lactate and glucose concentrations of the effluent. The runs consisted of eight low dose CVVHs with citrate, four high dose CVVHs with citrate, and seven low dose CVVHs without citrate. The CVVH settings during the different runs of CVVH are presented in Table 2. The patient characteristics can be found in Appendix A.

### 3.1. Bioenergetic Balance

The absolute and relative bioenergetic balances are depicted in Table 3. The mean absolute bioenergetic balance (over 24 h) during the low dose CVVH with citrate was 498 ± 110 kcal ranging from 339 kcal to 681 kcal. During the high dose CVVH with citrate, the mean absolute bioenergetic balance was 262 ± 222 kcal with a range of 56 kcal to 565 kcal, which was a statistically significant difference (*p* = 0.030) compared to the low dose CVVH with citrate. The mean absolute bioenergetic balance during the low dose CVVH without citrate was −189 ± 77 kcal with a range of −298 kcal to −92 kcal, which was a statistically significant difference compared to the low dose CVVH with citrate (*p* < 0.0001) and the high dose CVVH with citrate (*p* = 0.001).

The mean relative bioenergetic balance during the low dose CVVH with citrate compared to the true REE was 26 ± 9%, with a range of 14% to 42%. This was not a statistically significant difference from the high dose CVVH with citrate (*p* = 0.128), as the mean relative bioenergetic balance was 17 ± 11% compared to the true REE, with a range of 7% to 32%. During the low dose CVVH without citrate, the mean relative bioenergetic balance compared to the true REE was −13 ± 8%, with a range of −28% to −5 %, which was a statistically significant difference (*p* < 0.0001 and *p* = 0.001, respectively) compared to the two previous settings.

The adaptation for nutrition therapy ranges between −298 kcal and up to 681 kcal per day.

### 3.2. Non-Intentional Calories

Table 4 shows the exchange, due to the CVVH, of the different non-intentional calorie-containing molecules: citrate, glucose, and lactate. The relative energetic balance of the different molecules compared to the true REE can be found in Figure 1.

## 4. Discussion

The current study shows a large variability of bioenergetic balance during CVVH. Although in CRRT modes other than CVVH a similar effect has been described, to our knowledge this is a new finding [11,12]. The lowest bioenergetic balance was noticed during the CVVH without citrate, which correlates with previous findings, although there, lactate was used in the dialysis fluids [11]. Citrate is the most dominant contributor to positive bioenergetic balances (Figure 1).

Furthermore, citrate increases the REE [15]. Currently used predicting equations of REE do not take this change into consideration. Therefore, the predictive efforts without IC have failed to reproduce the individual REE systematically and accurately in intensive care unit (ICU) patients [19,20]. The MECCIAS-trial showed that during the CVVH, IC could measure the REE and, subsequently, could objectify the effect of citrate during the CVVH on individual metabolism [15]. By comparing the true REE measured by IC with the bioenergetic balance in the CVVH, we introduce a new concept. The MECCIAS-trial showed that citrate use during CVVH induced a raise in the REE of 330 kcal or approximatively 20% compared to the CVVH without citrate [15]. We found that the low dose CVVH with citrate delivers 40% of the calories needed compared to the true REE. The increase in calory delivery is more than the increase in REE. There would still be an overload of calories, which would largely overshoot the altered REE due to the citrate itself. Therefore, measuring the bioenergetic balance is necessary to avoid overfeeding.

Considerable variability in bioenergetic balances during CVVH with citrate was observed. The first explanation for this variability is the differences in the CVVH settings. Although the low dose CVVH with citrate induced the highest bioenergetic balance compared to the high dose CVVH with citrate, the delivery of citrate was clinically insignificantly different during the low dose CVVH versus the high dose CVVH. The difference in high or low dose did not contribute to the differences in the citrate-induced higher bioenergetic balance. On the other hand, the effluent dose was higher during the high dose CVVH. Therefore, more citrate was filtrated during the high dose CVVH resulting in lower bioenergetic balances of citrate, and this can explain the higher bioenergetic balance in the low dose CVVH compared to the high dose CVVH.

A second explanation for the variability in bioenergetic balances during the CVVH with citrate is the adaptation of the CVVH settings and fluids to individual needs. Due to the patient-related and technique-related issues, the citrate delivery varies between individuals [10]. Personalized CVVH therapy with personalized citrate delivery and removal explains the considerable variability in bioenergetic balances. Subsequently, if a precise feeding strategy is aspired, the variability of the bioenergetic balance of CVVH compared and combined with the measured REE dictates an individual approach. By doing so, under-, and probably more frequently overfeeding will be avoided.

The other novel finding in this manuscript was the in vivo negative balances during the low dose CVVH without citrate. Negative bioenergetic balances have been described in a theoretical in vitro model evaluating CRRT without glucose or citrate [21]. This is in contrast with previous in vivo reports in which glucose and lactate were identified with a dominant role on the impact of positive bioenergetic balances during CVVH [11,12]. The earlier reported studies used glucose and lactate containing fluids during CVVH, whereas our dialysis protocol uses glucose and lactate-free solutions. More recent CVVH guidelines advise using lactate-free buffer solutions [10]. The dialysis protocol of the intensive care department at UZ Brussel also uses glucose-free dialysis fluids as stress-induced hyperglycemia is frequently encountered in the ICU [22]. During the low dose CVVH without citrate, glucose loss was the primary cause of negative bioenergetic balances. Lactate impacted the bioenergetic balances less. If caloric prescriptions were made without accounting for these losses of calories, it would leave the patient highly underfed and might impair prognosis [2,4,16].

Applying a fixed correction factor for the caloric prescription depending on the CVVH setting without calculating the individual bioenergetic balance seems a practical approach; however, this is difficult to implement in clinical practice because of the multitude of influences on adequacy [23,24]. If a patient individualized calculation would not be feasible, we could try to implement the knowledge of the bioenergetic balance presented in this paper: During the low dose CVVH with citrate, the mean bioenergetic balance was 26% with a minimum of 14 to 42%. Normocaloric feeding implies a delivery of 70–100% of the REE [16]. This means that if bioenergetic balances would not be calculated, to avoid over- and underfeeding in all of the subjects, exactly 56% (56% + 14% = 70%) to 58% (58% + 42% = 100%) of the true REE should be delivered as calories to the patients. During the high dose CVVH with citrate, 63% to 68% of the true REE should be delivered as calories. During the low dose CVVH without citrate, the caloric prescription should be 105% to 98% of the true REE. Translation of such a narrow caloric target in clinical practice is practically impossible because of feeding challenges. These small margins would leave no room for errors in clinical practice where a lot of interfering factors can, but not necessarily will, influence the adequacy of the delivered nutrients, and thus, the calories [23,24]. A larger portion of the patients would be under- or overfed if such correction factors were applied. It seems preferable to measure the caloric balances and adapt accordingly.

Another consideration in CVVH is the loss of proteins in the effluent [25]. In the original MECCIAS trial, no information on protein losses were recorded as it was out of the scope. Although the energetic load of delivered proteins is added to the caloric delivery during nutrition therapy, the extra amount that should be given is to compensate for the losses. Therefore, it would not impact the caloric prescription as it is merely a supplementation [25]. However, further research integrating all of these different nutrition factors to optimize nutrition in CVVH patients seems necessary.

## 5. Conclusions

The bioenergetic balance of CVVH has considerable variation and can range from −28% up to +42% of the true REE. During CVVH with citrate, and predilution and postdilution without glucose or lactate, citrate is the predominant source of non-intentional calories. The absence of glucose in dialysis fluid contributes to negativizing the bioenergetic balance. Lactate has a minor impact on the bioenergetic balance during CVVH. When formulating a caloric prescription in a CVVH patient, an individual approach considering the impact of these non-intentional calories upon the bioenergetic balance is warranted.

## Figures and Tables

**Figure 1 nutrients-14-02112-f001:**
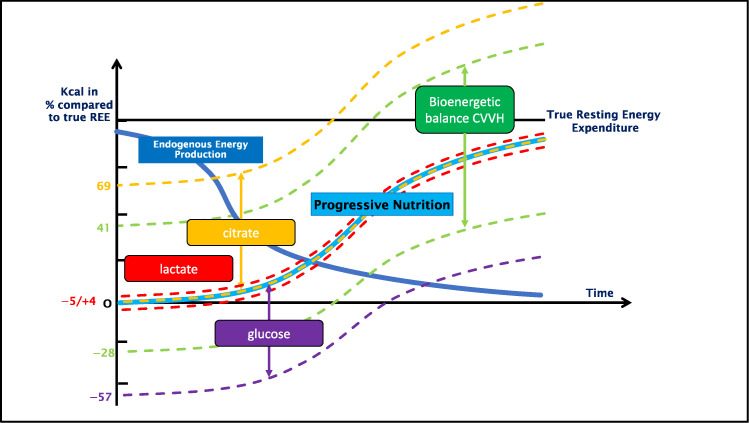
Bioenergetic balance of the CVVH compared to the true REE in relation to the ideal caloric delivery over time during progressive nutrition. This figure depicts the exchange of citrate, lactate and glucose in percentages compared to the true REE. It is placed against the need for progressive nutrition therapy, as recommended by the ESPEN guidelines [16], in relation to the endogenous energy production due to catabolism. The graph follows the recommendation of progressive nutrition therapy, illustrating the impact on total caloric load if the exchange of the different nutrients was not taken into consideration.

**Table 1 nutrients-14-02112-t001:** Composition of different dialysis fluids.

	Prismocitrate ^®^ 18/0	Prismocal B22 ^®^	Biphozyl ^®^	NaCl 0.9%
Na (mmol/L)	140	140	140	154
K (mmol/L)	0	4	4	0
Cl (mmol/L)	86	120.5	122	154
Mg (mmol/L	0	0	0.75	0
P (mmol/L)	0	0	1–2	0
HCO_3_ (mmol/L)	0	22	22	0
Citrate (mmol/L)	18	0	0	0
Glucose (mmol/L)	0	6.1	0	0
Lactate (mmol/L)	0	3	0	0

*^®^*: registered trademark.

**Table 2 nutrients-14-02112-t002:** Continuous venovenous hemofiltration (CVVH) settings.

	Low Dose CVVH with Citrate	High Dose CVVH with Citrate	Low Dose CVVH without Citrate
*n* =	8	4	7
Blood flow (mL/min)	150 ± 0	150 ± 0	150 ± 0
Predilution flow (mL/h)	1756 ± 264	1700 ± 147	1721 ± 296
Postdilution flow	506 ± 431	2300 ± 1036	464 ± 490
Postdilution fluid (n)			
− Biphozyl ^®^	1	0	1
− Prismocal 22 ^®^	3	2	3
− NaCl 0.9%	4	2	3
Effluent flow (mL/h)	2363 ± 476	4075 ± 974	2279 ± 551

Values are expressed in mean ± standard deviation; ^®^: registered trademark. CVVH: continuous venovenous hemofiltration.

**Table 3 nutrients-14-02112-t003:** Absolute and relative bioenergetic balances during the different Continuous venovenous hemofiltration (CVVH) settings.

		Low Dose CVVH with Citrate	High Dose CVVH with Citrate	Low Dose CVVH without Citrate
Absolute bioenergetic balance (kcal/day)	Mean	498 ± 110	262 ± 222	−189 ± 77
	Range	339 to 681	56 to 565	−298 to −92
Relative bioenergetic balance (%)	Mean	26 ± 9	17 ± 11	−13 ± 8
	Range	14 to 42	7 to 32	−28 to −5

Mean values are expressed with their standard deviation (±).

**Table 4 nutrients-14-02112-t004:** Exchange of non-intentional calories during the different Continuous venovenous hemofiltration (CVVH) settings.

		Low Dose CVVH with Citrate	High Dose CVVH with Citrate	Low Dose CVVH without Citrate	*p*-Value
Gain due to dialysis fluid of non-intentional caloric containing molecules	Citrate (mmol/24 h)	759 ± 114	734 ± 64	0	<0.001
	Glucose (g/24 h)	6 ± 14	38 ± 45	7 ± 15	0.083
	Lactate (mmol/24 h)	16 ± 38	104 ± 124	19 ± 40	0.083
Loss in effluent of non-intentional caloric containing molecules	Citrate (mmol/24 h)	168 ± 47	281 ± 73	0	<0.001
	Glucose (g/24 h)	64 ± 28	107 ± 40	57 ± 22	0.032
	Lactate (mmol/24 h)	64 ± 34	127 ± 74	60 ± 38	0.070
Total balance of non-intentional caloric containing molecules	Citrate (mmol/24 h)	591 ± 81	453 ± 60	0	<0.001
	Glucose (g/24 h)	−59 ± 24	−69 ± 53	−50 ± 20	0.607
	Lactate (mmol/24 h)	−48 ± 16	−22 ± 84	−42 ± 14	0.567
Absolute caloric balance (kcal/day)	Citrate	736 ± 101	564 ± 75	0	<0.001
	Glucose	−222 ± 90	−262 ± 202	−187 ± 74	0.584
	Lactate	−16 ± 5	−7 ± 27	3 ± 15	0.032
Relative caloric balance compared to the true Resting Energy Expenditure (REE) (%)	Citrate	40 ± 14% (26 to 69%)	44 ± 16% (34 to 69%)	0%	<0.001
	Glucose	−12 ± 7% (−25 to −5%)	−24 ± 24% (−57 to 0%)	−13 ± 8% (−28 to −5%)	0.300
	Lactate	−1 ± 1% (−2 to 0%)	−1 ± 3% (−5 to 1%)	0 ± 1% (−1 to 3%)	0.200

Values are expressed in mean ± standard deviation. In the relative caloric balance, the minimum and maximum were also added between ( ).

## Data Availability

Raw data can be obtained upon written request to the corresponding author.

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
