# Peer review of "Bioenergetic Balance of Continuous Venovenous Hemofiltration, a Retrospective Analysis"

_nutrients, 2022, doi:10.3390/nu14102112_

Round 1
Reviewer 1 Report
The authors estimated the effect of different hemofiltration solutions (containing citrate, glucose and lactate) on calorie intake during 19 hemofiltration sessions. These “non-intentional” calories were furthermore expressed as percentage of resting energy expenditure (REE) and could make up at maximum an additional 42% of REE or could lead to a loss of at max. 28% of REE. Citrate use (n=12) led to an increase in calories whereas citrate free hemofiltration led to a waste of glucose.
Major:
- In the abstract (line 29) the authors state that “These absolute and relative balances were statistically significant different from each other…” However Low and high dose citrate were not different when expressed as percentage of REE as later explained (line 123). This suggests that the patients had different REE making the difference in absolute caloric balance of lesser importance.
Minor:
- Table 2: The headings should be Low dose CVVH with citrate, High dose CVVH with citrate and Low dose CVVH without citrate. The same applies to table 3.
- Line 122: The sentence should be changed to “The mean … balance… during low dose… was 26+-9%.”
- The bioenergetic balance is calculated in the text (e.g. line 116 – 498+-110 Kcal). These numbers can be calculated from table 3 and should be added to table 3.
- Figure 1: What is depicted on the X-axis? Is it absolute caloric intake?
Author Response
The authors estimated the effect of different hemofiltration solutions (containing citrate, glucose and lactate) on calorie intake during 19 hemofiltration sessions. These “non-intentional” calories were furthermore expressed as percentage of resting energy expenditure (REE) and could make up at maximum an additional 42% of REE or could lead to a loss of at max. 28% of REE. Citrate use (n=12) led to an increase in calories whereas citrate free hemofiltration led to a waste of glucose.This is correct.
Major:
- In the abstract (line 29) the authors state that “These absolute and relative balances were statistically significant different from each other…” However Low and high dose citrate were not different when expressed as percentage of REE as later explained (line 123). This suggests that the patients had different REE making the difference in absolute caloric balance of lesser importance. Thank you, this sentence was removed from the abstract.
Minor:
- Table 2: The headings should be Low dose CVVH with citrate, High dose CVVH with citrate and Low dose CVVH without citrate. The same applies to table 3. This was altered
- Line 122: The sentence should be changed to “The mean … balance… during low dose… was 26+-9%.” Thank you, this was altered
- The bioenergetic balance is calculated in the text (e.g. line 116 – 498+-110 Kcal). These numbers can be calculated from table 3 and should be added to table 3. Thank you for this comment. We agree that it can be calculated from the different components. However, we feel that it is a different subject in a different paragraph. We made a separate table.
- Figure 1: What is depicted on the X-axis? Is it absolute caloric intake? It is time. We altered it in the manuscript.
Reviewer 2 Report
In this manuscript, Jonckheer and Colleagues report the bioenergetics balance of different CVVH settings due to citrate, glucose and lactate presence in the dialysis fluid. It is difficult to get the message of this paper. Besides showing the different molecules differently contribute to the bioenergetics balance during CVVH, the large variability of this contribution (-28 to +42%) makes it difficult to adapt nutrition solely on the basis of theoretical calculations. In order to make the manuscript more interesting, I would suggest giving practical clinical suggestions on the basis of the results of this work. For instance, for what I understood, both settings with citrate give a positive bioenergetics balance compared to “true REE” (14-42% for low dose CVVH+citrate and 7-32% for high dose CVVH+citrate) while low dose CVVH without citrate shows a negative bioenergetics balance compared to “true REE” in the order of -28 to -5%. Thus, could the authors suggest, as a rule of thumb, decreasing the caloric prescription for nutritional therapy of about 20% in case of CVVH with citrate while increasing caloric prescription of about 15% in case of CVVH without citrate? I think this is the essential point of the paper and should be extensively discussed and results better presented (maybe including a table in the manuscript to highlight these results that, in the present version of the manuscript are only poorly presented in the main text).
Other points:
- The manuscript should be revised to correct English language mistakes and rephrase some sentences that are not clear (as an example, but not limited to: “CRRT removes a part in the effluent [8,9]”; “The delivery of non-intentional calories can go up to 1434 kcal /day due to citrate, glucose, and lactate during CVVHDF [9,10]”; “The mean bioenergetic balance compared to the “true REE” was during low dose CVVH with citrate 26±9%, with a range of 14% to 42%, which was not statistically significantly different from high dose CVVH (p=0,128) as the mean bioenergetic balance was 17±11% compared to “true REE” with a range of 7% to +32%”; “The lowest bioenergetic balance is seen in CVVH without citrate and correlates with Balik et al. [9]”; “In CVVH with citrate predilution, glucose, and lactate-204 free solutions in post-dilution, citrate is the predominant source of non-intentional calo-205 ries”).
- In general, I would suggest more clarity in presenting and discussing the results as the manuscript is quite difficult to read and follow.
- Abstract: please, define ERR before its first use.
- Please, include in the Methods how glucose, lactate and citrate were measured.
- Please, define low and high dose CVVH in the Methods.
- Table 2: please be consistent throughout the manuscript with the terminology (i.e. “CVVH with citrate” should be replaced by “Low dose CVVH with citrate”, the same for CVVH without citrate)
- Why using the italic for the CVVH setting? Please check the entire manuscript and correct.
- How many patients performed the 19 CVVH sessions included in this study? What were their characteristics?
Author Response
In this manuscript, Jonckheer and Colleagues report the bioenergetics balance of different CVVH settings due to citrate, glucose and lactate presence in the dialysis fluid. It is difficult to get the message of this paper. Besides showing the different molecules differently contribute to the bioenergetics balance during CVVH, the large variability of this contribution (-28 to +42%) makes it difficult to adapt nutrition solely on the basis of theoretical calculations. In order to make the manuscript more interesting, I would suggest giving practical clinical suggestions on the basis of the results of this work. For instance, for what I understood, both settings with citrate give a positive bioenergetics balance compared to “true REE” (14-42% for low dose CVVH+citrate and 7-32% for high dose CVVH+citrate) while low dose CVVH without citrate shows a negative bioenergetics balance compared to “true REE” in the order of -28 to -5%. Thus, could the authors suggest, as a rule of thumb, decreasing the caloric prescription for nutritional therapy of about 20% in case of CVVH with citrate while increasing caloric prescription of about 15% in case of CVVH without citrate? I think this is the essential point of the paper and should be extensively discussed and results better presented (maybe including a table in the manuscript to highlight these results that, in the present version of the manuscript are only poorly presented in the main text). A table was added for clarity.
We added a paragraph with the positive and negative aspects of using a simple rule of thumb to modify caloric prescription depending on CVVH settings. We explained why we think it would lead to more over- and underfeeding and thus probably worsen outcome.
In our clinical practice we built such a calculator in excel where we input the different CVVH settings and 24h glycemia and lactatemia. Our dieticians are happy with this approach and acknowledge that the workload is minimal.
Other points:
- The manuscript should be revised to correct English language mistakes and rephrase some sentences that are not clear (as an example, but not limited to: The manuscript was revised to remove the spelling mistakes and rephrase some sentences (not only limited to the remarks below). Please see the tracked changes version.
- “CRRT removes a part in the effluent [8,9]”; This was altered
- “The delivery of non-intentional calories can go up to 1434 kcal /day due to citrate, glucose, and lactate during CVVHDF [9,10]”; This was altered
- “The mean bioenergetic balance compared to the “true REE” was during low dose CVVH with citrate 26±9%, with a range of 14% to 42%, which was not statistically significantly different from high dose CVVH (p=0,128) as the mean bioenergetic balance was 17±11% compared to “true REE” with a range of 7% to +32%”; This was altered
- “The lowest bioenergetic balance is seen in CVVH without citrate and correlates with Balik et al. [9]”; This was changed
- “In CVVH with citrate predilution, glucose, and lactate-204 free solutions in post-dilution, citrate is the predominant source of non-intentional calo-205 ries”). This was changed
- In general, I would suggest more clarity in presenting and discussing the results as the manuscript is quite difficult to read and follow. We presented our main results in a separate table (table 3) to add some clarity. We changed parts of the introduction and of the discussion.
- Abstract: please, define ERR before its first use. This was done
- Please, include in the Methods how glucose, lactate and citrate were measured. This was done
- Please, define low and high dose CVVH in the Methods. This information was added
- Table 2: please be consistent throughout the manuscript with the terminology (i.e. “CVVH with citrate” should be replaced by “Low dose CVVH with citrate”, the same for CVVH without citrate) This was altered in the whole manuscript.
- Why using the italic for the CVVH setting? Please check the entire manuscript and correct. This was done
- How many patients performed the 19 CVVH sessions included in this study? What were their characteristics? A table was added as an appendix.
Round 2
Reviewer 2 Report
The manuscript has improved, although it still requires a major language revision.
I would suggest making the Excel calculator available as Supplementary material.
Author Response
Dear reviewer,
please see attachment.
with kind regards,
Joop Jonckheer
